# A Review on Bioengineering the Bovine Mammary Gland: The Role of the Extracellular Matrix and Reconstruction Prospects

**DOI:** 10.3390/bioengineering12050501

**Published:** 2025-05-09

**Authors:** Fernando Chissico Júnior, Thamires Santos da Silva, Flávio Vieira Meirelles, Paulo Sérgio Monzani, Lucas Fornari Laurindo, Sandra Maria Barbalho, Maria Angélica Miglino

**Affiliations:** 1Department of Surgery, School of Veterinary Medicine and Animal Science, Universidade de São Paulo (USP), São Paulo 05508-270, SP, Brazil; fernandochissico@usp.br (F.C.J.); thamiresssilva@usp.br (T.S.d.S.); 2Department of Veterinary Medicine and Animal Science, School of Veterinary Medicine and Animal Science, Universidade Save (UniSave), Chongoene 1200, Mozambique; 3Department of Veterinary Medicine, Faculty of Animal Science and Food Engineering, Universidade de São Paulo (USP), Pirassununga Campus, Pirassununga 13635-900, SP, Brazil; meirellf@usp.br; 4Department of Food Engineering, Faculty of Animal Science and Food Engineering, Universidade de São Paulo (USP), Pirassununga Campus, Pirassununga 13635-900, SP, Brazil; monzani.paulo@gmail.com; 5Department of Biochemistry and Pharmacology, School of Medicine, Universidade de Marília (UNIMAR), Marília 17525-902, SP, Brazil or lucaslaurindo@unimar.br (L.F.L.); smbarbalho@gmail.com (S.M.B.); 6Postgraduate Program in Structural and Functional Interactions in Rehabilitation, School of Medicine, Universidade de Marília (UNIMAR), Marília 17525-902, SP, Brazil; 7Postgraduate Program in Animal Health, Production and Environment, School of Veterinary Medicine, Universidade de Marília (UNIMAR), Marília 17525-902, SP, Brazil; 8Department of Animal Anatomy, School of Veterinary Medicine, Universidade de Marília (UNIMAR), Marília 17525-902, SP, Brazil

**Keywords:** mammary gland, decellularization, recellularization, extracellular matrix

## Abstract

The mammary gland is a modified sweat gland responsible for milk production. It is affected by diseases that reduce animals’ quality of life, consequently leading to economic losses in livestock. With advancements in tissue bioengineering and regenerative medicine, studying the extracellular matrix (ECM) of the bovine mammary gland can improve our understanding of its physiology and the processes that affect it. This knowledge could also enable the development of sustainable therapeutic alternatives for both the dairy production chain and human oncology research. A common approach in regenerative medicine is decellularization, a process that removes all cells from tissue while preserving its architecture and ECM components for subsequent recellularization. The success of recellularization depends on obtaining immunologically compatible scaffolds and using appropriate cell culture sources and methods to ensure tissue functionality. However, tissue culture technology still faces challenges due to specific requirements and high costs. Here, we review the literature on biomaterials and tissue engineering, providing an overview of the ECM of the bovine mammary gland and advances in its bioengineering, with a focus on regenerative medicine for bovine species. The methodology employed consists of a structured search of scientific databases, including PubMed, Google Scholar, and SciELO, using specific keywords related to tissue engineering and the bovine mammary gland. The selection criteria prioritized peer-reviewed articles published between 2002 and 2025 that demonstrated scientific relevance and contributed to the understanding of bovine mammary gland bioengineering. Although research on this topic has advanced, vascularization, tissue maturation, and scalability remain key barriers to widespread application and economic viability.

## 1. Introduction

Dairy farming holds a strategic position within the global food industry, with bovine milk representing one of the primary sources of nutrition for billions of people worldwide [1,2]. However, the sector faces persistent challenges related to increasing production to meet the growing population demand, improving milk quality, and controlling diseases that negatively impact animal health and productivity, particularly bovine mastitis [2,3,4,5].

Recognized as the most prevalent disease in dairy herds, mastitis is also one of the leading causes of economic losses in the dairy sector [6,7]. The integrity and functional efficiency of the mammary gland are critical determinants of productive performance, directly influencing the volume and quality of milk produced [1,2,5]. The disease may be caused by various etiological agents, including bacteria, fungi, yeasts, and viruses, as well as non-infectious factors such as stress and physical trauma [4,8]. Mastitis leads to decreased milk secretion and disrupts the permeability of the blood–milk barrier [6,9].

Clinically, mastitis presents in two forms: clinical and subclinical. The subclinical form, which is more prevalent, is asymptomatic and difficult to diagnose, being responsible for a significant portion of economic losses due to decreased milk yield and quality [6,9,10,11]. The clinical form is characterized by visible mammary gland inflammation, requiring therapeutic intervention and leading to milk disposal and reduced reproductive efficiency [6,9,12]. Both forms represent a global challenge to dairy production, with economic and public health implications, particularly in regions where raw milk is consumed daily, increasing the risk of zoonotic transmission [4,13,14].

Numerous microorganisms have been identified as causative agents of mastitis, with *Staphylococcus aureus*, *Staphylococcus dysgalactiae*, *Streptococcus uberis*, *Streptococcus agalactiae*, and *Escherichia coli* being frequently associated with both clinical and subclinical infections [15]. In addition, other opportunistic pathogens, such as *Pseudomonas* spp., *Mycoplasma* spp., various yeasts, and *Nocardia* spp., may also be involved in the disease’s etiology [4,6,9,16,17].

Studies investigating its biology are essential, given the importance of the mammary gland in milk production. In this context, in vitro reconstruction of structures resembling functional mammary tissue poses a significant challenge, particularly in livestock. If successful, this technique could enhance our understanding of the physiological processes of milk secretion and the pathophysiological events resulting from viral or bacterial infections [4,18]. Furthermore, this knowledge could lead to significant advances in regenerative medicine, breast cancer research, understanding the mechanisms involved in milk production, and developing dairy products for the dairy industry [18,19], contributing to various research.

In this context, several three-dimensional (3D) cell culture studies have been conducted to develop organs in vitro [18,19,20]. Among the promising approaches in tissue bioengineering, the decellularized extracellular matrix (dECM) stands out, as it promotes cell proliferation, migration, and differentiation [21]. The dECM is created by removing cellular components from the organ while preserving extracellular matrix (ECM) components. It is an ideal biomaterial for mimicking natural tissue due to its structure and biomechanical properties, similar to those of the original organ. It also provides the necessary chemical and physical signals to support new cells and form compatible tissue [22,23,24] (Figure 1).

In addition, the use of 3D printing in tissue bioengineering can revolutionize the study of the ECM of the bovine mammary gland by enabling the creation of biomimetic models that accurately replicate its architecture and composition [24] (Figure 1). These advancements are particularly relevant to the dairy sector, which faces challenges related to milk production efficiency and cost reduction across the production chain, especially concerning the treatment of diseases that compromise mammary gland health [25]. By facilitating the in vitro production of milk and bioactive proteins, 3D bioprinting provides sustainable and efficient alternatives to traditional food production systems, which are often constrained by adverse climatic conditions and limited investment [26].

Considering that the components and architecture of the dECM are essential for mimicking the microenvironment of tissues cultured in vitro, this review aims to provide an overview of the mammary gland and studies on the decellularization and recellularization of this organ.

## 2. Macro and Microscopic Characterization of the Bovine Mammary Gland

### 2.1. The Anatomy of the Bovine Mammary Gland

Embryonic development of the mammary gland in cattle begins around the 35th day of gestation, forming the mammary line, also called the milk line or mammary ridge. This structure corresponds to a thickening of the ectoderm, located on the ventral region of the embryonic abdominal wall and arranged parallel to the ventral midline of the trunk. Evidence indicates that its formation occurs independently of systemic hormonal stimuli and is primarily regulated by signals originating from the underlying mammary mesenchyme [27]. From the mammary line, mammary buds develop through localized ectoderm thickenings, representing the future sites of mammary gland development. Around the 60th day of gestation, the mammary bud invaginates into the dermis, a process characterized by the proliferation and penetration of ectodermal cells into the underlying mesenchyme, giving rise to the primary mammary bud, or placode [1,28]. It is noteworthy that teat formation coincides with the invagination and proliferation of the mammary bud between the 60th and 65th days of gestation, resulting in the development of four distinct teats. The development of glandular tissue accelerates between the 5th and 6th months of gestation and is characterized by the expansion of the gland cistern and the branching of ducts and terminal buds. Duct formation is a fundamental process in the development of the bovine mammary gland. It begins around the 100th day of gestation, at the proximal end of the primary mammary bud, and progressively advances toward the distal end, ultimately creating an opening to the exterior of the teat. This ensures a functional connection to the external environment, which is essential for efficient milk transport after the onset of lactation. Secondary mammary buds emerge from the proximal region of the primary bud, and their cavities give rise to the central lactiferous ducts. Approximately by the sixth month of gestation, most of the principal components of the mammary gland—including the four glands, the median suspensory ligament, the teats, and the gland cistern—are already formed. This process is regulated by specific signaling molecules and cellular interactions that facilitate the formation of lumens within the epithelial cords [1].

It is composed of a teat, covered by glabrous skin, and a glandular body, covered with hair. Anatomically, the bovine mammary gland is divided into four quarters, each consisting of tissues that support, produce, and store milk [5,29] (Figure 2).

### 2.2. Histological Characterization

Histologically, the bovine mammary gland is a modified exocrine sweat gland of the tubule–acinar-type compound, separated by connective tissue. Its structure varies with hormonal changes throughout the estrous cycle [29,30].

The glandular structure comprises two distinct tissue types: the parenchyma and the stroma. The parenchyma, or glandular tissue, comprises luminal cells organized into alveoli, spherical structures that produce milk. These alveoli aggregate to form multi-alveolar mammary lobules organized within a branched ductal network and contain adipose tissue within the parenchyma [30,31,32,33]. Surrounding the alveoli are basal myoepithelial cells and cuboidal epithelial cells, forming the outer layer of the gland [29,32,34]. These two cell types are anchored to a collagen-rich basal membrane.

The secretory portion of the gland (*pars glandularis*) consists of glandular epithelial and connective tissue, separated by connective tissue septa that form the glandular sinus [29]. Alveoli are the morphofunctional units of the mammary gland, responsible for milk synthesis and secretion. They consist of secretory epithelial cells that proliferate into milk-producing cells. Alveoli are lined with simple cuboidal or isoprismatic epithelia, with cell height varying according to functional state; they are shorter when the alveolar lumen is filled [30,32]. Lobular activity varies throughout the secretory cycle, leading to milk production variations and secretory cell morphology changes within the same lobule [32,33,34].

The mammary stroma is a complex tissue composed of the ECM and various cell types that maintain the structural integrity of the gland and regulate epithelial metabolism. This stroma consists of loose connective tissue, which divides the parenchyma into multiple alveolar lobules, capillaries, fibroblasts, myoepithelial cells, and reticular fibers, forming the intraparenchymal interstitial connective tissue [30,32,33,34]. The mammary stroma also contains adipocytes, immune cells, and nerves [31,35].

In the basal region of the glandular sinus, several lactiferous ducts open, lined with stratified squamous epithelia. As these ducts extend toward the lobules, the epithelium thins into two layers of cylindrical cells, transitioning to simple cuboidal epithelia near the alveoli. The lobular and lobar ducts contain longitudinally arranged smooth muscle cells [33] (Figure 2B).

### 2.3. The Extracellular Matrix of the Mammary Gland

The ECM is a 3D, acellular structure that occupies the extracellular space and supports the cells of tissues and solid organs [36]. Synthesized during embryonic development, the ECM interacts with tissue cells to modulate the microenvironment, facilitating cell migration, proliferation, differentiation, and homeostasis [37,38,39]. The ECM is a highly dynamic system that continuously remodels its molecular components to endow organs with specific biochemical and mechanical properties, such as elasticity, tensile strength, and resistance to compression [39].

The ECM comprises over 300 proteins, including collagen, elastin, glycosaminoglycans, fibronectin (FN), laminin (LN), and other glycoproteins [39,40]. These proteins provide structural support and serve as reservoirs for growth factors and bioactive molecules, essential for regulating cellular behavior and tissue-specific functions. Collagen and elastin, in particular, are crucial for maintaining the osmotic pressure of the tissue and regulating intracellular signaling cascades, whereas glycosaminoglycans serve as scaffolds for transporting growth factors [35,36].

In the mammary gland, the ECM consists of the basement membrane, which separates epithelial and stromal layers, and the interstitial matrix, which surrounds cells and forms a porous, 3D scaffold. When decellularized, this structure is a scaffold suitable for biomedical applications [41]. Beyond providing mechanical support, the ECM modulates cellular functions by facilitating intercellular and intracellular signaling, thereby contributing to proliferation, differentiation, migration, and apoptosis. These processes are mediated by bioactive molecular factors that enable cell-to-cell and cell-to-matrix communication [42]. Although the mammary ECM shares similarities with other tissues, its specific properties ensure mechanical support, tissue homeostasis, and the delivery of growth factors and cytokines, which collectively confer tensile strength to the tissue and maintain functional integrity. As such, the ECM is a critical modulator of cellular functions [43].

Overview and assembly of the mammary gland ECM are performed during intrauterine development, where mutual interactions occur between embryonic cells and epithelial cells, myoepithelial cells, fibroblasts, adipocytes, endothelial cells, and immune cells that communicate bidirectionally with cells through the support of the ECM, while we build, model, and remodel the ECM [37,44,45].

The components of the mammary gland ECM will be described based on three major structural categories: the highly specialized basal lamina that directly abuts the basal side of the mammary epithelial cells, the intra and interlobular stroma adjacent to the alveoli and lobules, respectively, and the fibrous connective tissue that is devoid of any epithelium [44]. Table 1 presents the primary components of the mammary gland ECM, highlighting its structure and applications.

### 2.4. Comparative Analysis of the Extracellular Matrix of the Mammary Gland Between Cattle and Small Domestic Ruminants

The basement membrane of the bovine mammary gland plays an essential role in the mammary tissue’s structural and functional organization. It comprises two main compartments: one directly supporting the mammary epithelium and the interstitial stroma, providing connective support [54].

Comparisons between the mammary ECM of cattle and small ruminants may reveal distinct biochemical and biomechanical characteristics, with relevant implications for tissue bioengineering. Table 2 compares the ECM components of the mammary gland in bovines and small ruminants.

### 2.5. The Role of the Extracellular Matrix in the Immune Response During Mastitis

The ECM, particularly the basal membrane, is a physical barrier against disseminating pathogens in mammary tissue [61]. During mastitis, the integrity of this structure, composed mainly of collagen IV and LN, may be compromised, facilitating infectious spread. In addition to their structural function, these components interact with immune cells, modulating neutrophil responses [61,62].

The ECM also plays a key role in the recruitment and guidance of immune cells by binding to chemokines, which form chemotactic gradients that direct cell migration toward infection sites [61]. Neutrophil interaction with ECM components can enhance or suppress effector functions, such as producing reactive oxygen species, contributing to a more efficient immune response [61,63].

Proteins such as LN and FN influence immune cell adhesion and activation. Moreover, FN is exploited by pathogens such as *Staphylococcus aureus* and *Streptococcus* spp., which express fibronectin-binding proteins (FnBPs) that promote adhesion to epithelial cells and facilitate colonization of the mammary gland [61,63,64].

Other components, such as elastin, may contribute to tissue integrity preservation during inflammation and edema. Proteoglycans and glycosaminoglycans—such as hyaluronan and Perlecan—also influence immune responses by modulating immune cell activity [65,66].

During mastitis, the ECM undergoes extensive remodeling, including increased type I collagen and FN deposition, which may induce fibrosis and alter tissue architecture [61,66]. The heightened activity of matrix metalloproteinases (MMPs), especially MMP-9, contributes to ECM degradation, facilitating immune cell infiltration and exacerbating tissue damage. Changes in ECM stiffness resulting from these modifications can significantly influence immune cell behavior and disease progression [64].

### 2.6. Therapeutic Strategies for Bovine Mastitis

The most commonly employed therapeutic approach for treating bovine mastitis involves using antimicrobial agents, particularly beta-lactam antibiotics, such as penicillins and cephalosporins. However, the indiscriminate administration of these drugs has contributed to the emergence of antimicrobial resistance, which has become a global public health concern [8,67,68]. The selection of the appropriate therapeutic protocol depends on the severity of clinical signs and the definitive diagnosis [17,69].

Given the growing ineffectiveness of conventional antimicrobials, research has focused on developing non-invasive alternative therapeutic strategies that present a low risk of inducing microbial resistance. In this context, there is increasing interest in alternative therapies, including using plant extracts, essential oils, nanoparticles, antimicrobial peptides, and bacteriophages, which have shown efficacy against multidrug-resistant bacterial strains [14,70,71]. Furthermore, immune-modulating strategies, such as recombinant cytokines and specific immunizations, are being explored.

In parallel, advances in developing in vitro models of the bovine mammary gland have enabled a more detailed investigation of pathogen–host interactions and the preliminary assessment of new therapeutic molecules, thereby reducing the reliance on in vivo experimental models [72].

## 3. Decellularization Method for Obtaining Extracellular Matrix

The use of the ECM in tissue bioengineering requires decellularization, a process that removes cells and potential immunogenic factors that could compromise the scaffold’s biocompatibility [73]. The procedure involves cell membrane lysis, separation, and solubilization of cellular components, and subsequent removal of cellular debris [36,73]. After decellularization, tissues undergo washing protocols to remove residual chemicals [74].

Decellularization can be achieved through different methods, including natural, chemical, physical, or a combination [41]. The most effective approach integrates physical, enzymatic, and chemical processes [74]. Importantly, decellularization must preserve the structural, biochemical, and biomechanical properties of the ECM to replicate the native cellular microenvironment accurately [53].

Physical treatments include agitation, sonication, mechanical pressure, and freeze–thaw cycles [43], all of which lyse the cells and release intracellular contents [73]. Enzymatic methods (e.g., trypsin) and chemical treatments (e.g., ionic solutions and detergents) disrupt cell membranes and intra- and extracellular connections [75].

Among these methods, detergent-based approaches are the most used due to their effectiveness in lysing cell membranes and removing cellular components. However, detergents can compromise ECM integrity during recellularization. To mitigate this, optimizing the concentration of sodium dodecyl sulfate (SDS) and minimizing tissue exposure time is critical to preventing ECM disintegration and protein loss [73].

SDS is the most widely used detergent for decellularization [76]. However, selecting decellularization agents must consider tissue-specific factors, such as cellularity, density, lipid content, and thickness [77]. For solid organs, fragmenting the tissue into thin slices and placing them in a chemical bath has been suggested as an effective strategy. However, this approach may alter organ architecture and limit the use of dECM as an intact structure [78]. The primary decellularization methods are summarized in Table 3.

### 3.1. Analysis of Decellularization Efficiency

An essential aspect of validating tissue decellularization protocols is implementing quality control. This process ensures the efficacy of decellularization, eliminating potential interferences in the experimental results. The minimum criteria for adequate decellularization are based on the quantification of deoxyribonucleic acid (DNA) content and the absence of nuclear material in tissue sections stained with hematoxylin and eosin (H&E) [73,74]. The gold standard for evaluating decellularization efficiency includes the following: (1) DNA content lower than 50 ng/mg of dry weight; (2) DNA fragment length less than 200 base pairs; and (3) the absence of visible nuclear material in tissue sections stained with H&E or 4′,6-diamidino-2-phenylindole (DAPI) [91].

### 3.2. Ideal Properties of ECM

To achieve satisfactory results in tissue engineering, it is essential to use high-quality dECMs that exhibit optimal biocompatibility, biodegradability, and angiogenic potential [53]. Biocompatibility is a concept applied across various fields. Generally, it refers to the ability of a material to perform a specific function or elicit a desired response in a given application without causing adverse effects [39,92]. In tissue engineering, the biocompatibility of the dECM is characterized by its non-immunogenicity, minimal toxicity, and support for healthy cell adhesion, proliferation, and migration. These attributes are achieved through strong cell adhesion and supportive structural features [93,94].

Biodegradability in tissue engineering involves utilizing biodegradable materials to fabricate scaffolds that facilitate tissue regeneration. It is defined as the process by which biomaterials are solubilized in tissue fluids, reducing their initial volume or eventual disappearance at the implantation site over time. Controlled biomaterial degradation is critical for facilitating new tissues’ growth and proper integration [39,95].

The biodegradation of the ECM occurs through remodeling, which involves the breakdown and removal of old or damaged components via enzymatic activity [96]. This allows the ECM to renew itself by maintaining a balance between the synthesis and degradation of its components. This equilibrium is crucial for cellular and organ homeostasis. Two enzyme families regulate ECM remodeling: MMPs, which drive degradation, and lysyl oxidases (LOX), which are essential for post-translational modifications [97]. Additionally, external stimuli, such as transforming growth factor-beta (TGF-β), influence ECM remodeling, which enhances ECM production and upregulates genes related to its composition [91,98].

The angiogenic potential of the dECM refers to its ability to stimulate the formation of new blood vessels. Angiogenesis is essential for ensuring an adequate blood supply, oxygen, and nutrients to tissues and organs developed through tissue engineering [39]. Although establishing vascular networks in vivo using the dECM remains challenging, promising results have been observed in specific tissues, such as the skin, bladder, and cartilage [99].

## 4. Recellularization Method

Recellularization involves repopulating dECM scaffolds with organ-specific cells or stem cells to reconstruct microanatomy and restore organ-specific functions. This process requires creating an environment that mimics the physiological conditions of the target tissue [100,101]. Typically, recellularization occurs in static cell seeding and dynamic recellularization. The perfusion technique is essential for re-endothelialization of the vascular network and ensuring uniform cell distribution throughout the parenchyma. During dynamic recellularization, the scaffold is perfused with a cell suspension in a culture medium [101,102].

Successful recellularization depends on the use of appropriate cell sources, efficient culture methods, and, in some cases, the utilization of bioreactors. Organ recellularization is done by restoring the parenchyma and vascularization, and fully supporting the tissue components [75,103]. Two techniques are commonly employed, depending on the cell type: vascular network recellularization and direct cell injection into the parenchymal compartment of the organ. In some instances, both techniques are combined [102]. However, whole-organ recellularization technology is limited by tissue complexity, specificity, and high costs [39,75,77,104]. Various cell types are used for organ culture and recellularization, including embryonic stem cells, bone marrow-derived stem cells, adipose tissue-derived stem cells, mesenchymal stem cells (MSCs), induced pluripotent stem cells (iPSCs), human umbilical vein endothelial cells, small airway epithelial cells, pulmonary alveolar epithelial cells, microvascular endothelial cells, and alveolar epithelial type II cells [32,73].

### 4.1. Cell Types Used for the Recellularization of Parenchymal Organs

Recellularization of parenchymal organs relies on selecting appropriate cell types to restore functionality and mimic the organ’s native microenvironment [104]. Primary cells derived from the mammary gland, such as epithelial and myoepithelial cells, are commonly used due to their ability to replicate the tissue’s structural and functional characteristics [105].

Additionally, MSCs and iPSCs are increasingly utilized for their ability to differentiate into various cell types, including those necessary for mammary tissue regeneration [75,106,107]. Co-cultures, which combine multiple cell types such as fibroblasts and endothelial cells, are crucial for creating a supportive ECM and promoting vascularization. Cell sourcing and integration advancements are essential for achieving efficient recellularization and enabling bioengineered mammary tissue development for research and biotechnology applications [104,105]. Table 4 lists the central cells used in the recellularization process.

### 4.2. Support Cells

Integrating mural, stromal, immune, and interstitial cells during organ recellularization is critical for cellular renewal and adequate injury response. For instance, fibroblasts help remodel the ECM and enhance cell function within scaffolds [116,117,118]. Cultivation methods vary depending on the target organ and may involve perfusion, direct parenchymal injection, or a combination of both, as previously described. Although a vascularized network is shared with all organs, non-vascular routes can improve cell cultivation [75,110].

While the cellular scaffold provides the necessary structural foundation for cell adhesion and proliferation, bioengineered tissues’ proper functionality and development rely on controlled and optimized culture conditions [101,119]. In this context, bioreactors play a crucial role, as they are designed to replicate the specific physiological requirements of each organ or tissue [120]. These systems create a dynamic in vitro environment, supplying essential mechanical, chemical, and biological stimuli for the efficient recellularization of the bovine mammary gland, ensuring favorable conditions for cell growth, differentiation, and the formation of functional tissue [39,102,120]. Key monitored parameters include pH, pO_2_, pCO_2_, temperature, electrolyte levels, glucose and lactate concentrations, and perfusion-related metrics such as pressure and flow rate. The stability of these parameters is vital, particularly for long-term cultures [75,120].

## 5. Decellularization and Recellularization of the Mammary Gland

Many decellularized organs can recellularize, a finding that has contributed substantially to both experimental biology and preclinical applications. Common sources of dECMs include the liver, bladder, heart, respiratory tract, nervous tissue, adipose tissue, cartilage, blood vessels, skin, skeletal muscle, bone, tendons, ligaments, small intestine submucosa, and mammary glands [16,93,102,121].

Although studies on mammary gland decellularization and recellularization remain limited, research on canine and mammary organoid cultures in humans and mice shows promise for bioengineering mammary glands [29,122]. This field’s advances may enhance our understanding of mammary gland physiology and diseases, especially in bovines.

One study decellularized canine mammary glands using SDS for six days at 60 rpm, comparing scaffolds derived from healthy and tumor-bearing mammary glands. The ECM preserved ultrastructural components, including collagen fibers and vascular regions, with no nuclear content. During recellularization, fibroblasts and yolk sac cells (native) treated with vascular endothelial growth factor (VEGF) adhered and survived, but mammary tumor cells did not [29]. Another study on human and mouse mammary organoids showed morphological similarities to native tissue, with cell development and maturation occurring in a native-like environment of key components, such as type I collagen, hyaluronic acid, FN, and LN. These organoids replicated various hormonally regulated physiological and morphological events that occur in vivo, but could not fully mimic the complexity of native tissue [122]. These findings provide new insights into the physiological changes in bovine mammary glands. Organoids, miniature 3D-cultured versions of tissues and organs, emerge as a promising alternative to the scarcity of organs for in vivo experimentation.

Studies on human mammary organoids show that appropriate decellularization protocols preserve tissue integrity [122,123]. However, the architecture and development of human mammary glands differ from those of mice [32]. In human organoid cultures, ductal morphogenesis is regulated differently by the epidermal growth factor receptor (EGFR) and fibroblast growth factor receptor-2 (FGFR2) ligands. Although mouse organoids readily branch in response to FGFR2 ligands, human organoids rely on EGFR signaling for ductal growth and myoepithelial cell regulation [32,123]. Although branching morphogenesis is well studied in 3D cell cultures, the lactation-related functions of the mammary gland remain underexplored, and its complete in vitro regeneration remains unfeasible [19,20].

Culturing mammary cells in structures that mimic native tissue is crucial for maintaining their functionality. The functional assessment of these organoids involves detecting β-casein and other milk proteins using immunological techniques [124]. However, the mammary gland’s cellular diversity—comprising epithelial cells, fibroblasts, endothelial cells, and adipocytes—complicates efficient coculture within scaffolds [125]. In this context, microfabrication can aid in regenerating more complete structures. For example, combining adipose tissue engineering with microfabrication techniques has produced a 3D epithelium that mimics mammary gland ductal structures. Preadipocytes seeded in collagen gel formed cavities and differentiated, while epithelial cells covering these cavities developed into tubules [20]. Some examples of the application of decellularized and recellularized mammary glands are summarized in Table 5.

### 3D Printing Is Used for the Recellularization Process

3D printing has emerged as a groundbreaking technology in tissue bioengineering, creating highly customizable and precise structures for cell support and tissue recellularization [131,132,133,134]. Using advanced biomaterials, such as biocompatible hydrogels and biodegradable polymers, it is possible to fabricate 3D scaffolds that accurately replicate the architecture and mechanical properties of native tissues [135,136]. These scaffolds provide a structural framework for cell growth and allow for the incorporation of bioactive factors, such as adhesion proteins and chemical gradients, which are essential for cell differentiation and the development of functional tissues, thereby increasing the success of many studies [133,136].

In the recellularization process, 3D-printed scaffolds play a crucial role by creating customized microenvironments that support cell adhesion, proliferation, and organization. Moreover, 3D printing enables the integration of multiple cell types in specific patterns, facilitating the construction of complex tissues, such as blood vessels and epithelial interfaces [133,134,137,138]. Combined with technologies like dynamic bioreactors and cell-assisted bioprinting, this approach has expanded the possibilities for tissue regeneration, driving significant advances in developing functionally mature tissues for applications in transplantation, experimental models, and personalized medicine [132,139]. Despite its potential, challenges related to vascularization, scalability, and tissue maturation still need to be addressed for the complete clinical application of this technology [140]. Table 6 summarizes some studies that utilized 3D printing or 3D culture techniques in the recellularization of mammary gland tissues in various species.

## 6. Limitations of Extracellular Matrix Recellularization

Although the use of the dECM in tissue bioengineering offers numerous advantages, its recellularization poses challenges due to several specific factors described below.

Non-homogeneous cell distribution is a significant challenge in recellularization, as it can impair the functionality and efficacy of engineered tissues or organs [39,147]. Cell migration in 3D tissues is modulated by the balance between cellular deformability and physical constraints, which is regulated by ECM proteolytic enzymes and integrin–actomyosin interactions [148]. Studies have shown that ECM non-uniformity often results from nutrient gradients. To address this, microgels with radii smaller than 200 μm have demonstrated the ability to achieve nearly uniform ECM deposition, with initial cell density having minimal impact [39].

Cell adhesion and retention are dynamic and complex processes influenced by the composition, mechanical properties, and biochemical signals of the dECM, as well as cellular requirements and inhibitory molecules [39,45]. Beyond blood cells, other normal tissue cells also require anchorage to the ECM for proper function [149]. Cell–ECM interactions are crucial for maintaining a suitable cellular environment and ensuring tissue and organ homeostasis [150,151]. These interactions are mediated by organelles classified as focal adhesions and hemidesmosomes. Hemidesmosomes, which bind to intermediate filaments, are particularly prominent in epithelial tissues that connect to the basement membrane [39,152].

Mechanical integrity and stability are also essential aspects of tissue engineering, as scaffolds provide the necessary mechanical support and shape for tissues and organs. To be effective, the mechanical properties of scaffolds must closely match those of the host tissue [149]. These properties are determined by the composition of key ECM components, such as elastic fibers, fibrillar collagens, glycosaminoglycans, and proteoglycans, which influence cell–matrix interactions as well as the chemical composition and structural organization of the ECM [153].

Although the absence of cells in the dECM reduces immune reactions by removing immunogenic material, immunological challenges remain in clinical applications [39]. Decellularized ECM scaffolds are considered viable due to their biocompatibility and low immunogenicity. However, interactions between dECM scaffolds and the immune system are primarily mediated by damage-associated molecular patterns (DAMPs), which are released into the extracellular environment following tissue injury [130,151].

## 7. Future Perspectives

Tissue bioengineering of the bovine mammary gland represents a promising and innovative approach with the potential to transform in vitro milk production and support research into mammary gland physiology. The integration of advanced technologies—including 3D-printed biomaterials, biocompatible scaffolds, intelligent bioreactors, and artificial intelligence—is expected to improve bioengineered tissues’ structural and functional accuracy, enabling the sustainable production of milk proteins and bioactive compounds.

Beyond its relevance to the dairy and pharmaceutical industries, this field contributes to the broader landscape of regenerative medicine, with potential applications in organ biofabrication, drug testing platforms, and personalized cell-based therapies. However, vascularization, tissue maturation, and scalability challenges remain significant barriers to its widespread adoption and commercial viability.

Despite these limitations, ongoing advancements indicate that tissue bioengineering may offer sustainable and innovative alternatives to traditional practices, contributing to enhanced animal welfare, food security, and global health.

## Figures and Tables

**Figure 1 bioengineering-12-00501-f001:**
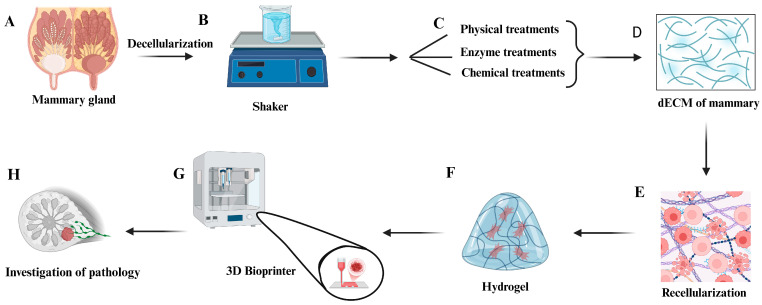
A graphic abstract of bovine mammary gland fragments’ decellularization and recellularization process. (**A**) The bovine mammary gland. The source for obtaining fragments to decellularize using the chemical sodium dodecyl sulfate (SDS) and physical methods (constant stirring). (**B**) A shaker with a flask containing mammary fragments in the decellularization process. (**C**) The main techniques used for tissue decellularization. (**D**) The decellularized extracellular matrix (dECM) mammary gland (scaffolds). (**E**) Recellularization to verify the cytocompatibility of the dECM for different cell lines. (**F**) A hydrogel. Injectable hydrogels stimulate recellularization. (**G**) A 3D bioprinter. The production of scaffolds with a 3D microstructure like mammary gland tissue, with the hydrogel produced for the dECM. (**H**) A bioprinted mammary gland model. Studies with a mammary model to investigate pathologies and the mechanisms of milk production. Created with BioRender (https://www.biorender.com/ (Accessed on 27 April 2025)).

**Figure 2 bioengineering-12-00501-f002:**
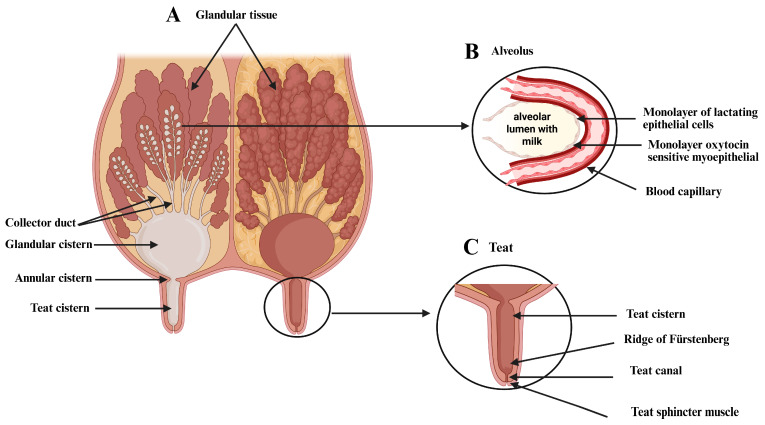
An image showing the macro and microstructure of the bovine mammary gland. (**A**) The glandular tissue, ducts, glandular cisterns, annular cistern, and teat cistern. (**B**) A cross-section of an alveolus. (**C**) A diagram of the anatomy of the teat and its components: teat cistern, ridge of Furstenberg, teat canal, and teat sphincter muscle. Created with BioRender (https://www.biorender.com/ (Accessed on 27 April 2025)).

**Table 1 bioengineering-12-00501-t001:** Components of mammary gland ECM.

Mammary Gland Structure	Components	Description	Applications	Reference
Basal lamina	Collagen IV	It is a network-forming class. It is the main component of the basal lamina of the mammary gland. It is a heterotrimer and composed of six possible α chains. It is also a significant basal lamina component and is considered its primary scaffold protein.	It supports the basal lamina structure during embryogenesis and in mammary epithelial cells.	[44]
	Nidogens	Mesodermally derived fibroblasts synthesize sulfated glycoproteins (150 kDa).	They produce Laminins (LN, Ln, Lm, or Lam), which are stabilizing components of the basal lamina. They allow for the connection between LN-111 and collagen IV.	[44]
Intra and interlobular stroma	Collagen I	Fibrillar collagen I is the main protein of the stroma of the mammary gland that supports the formation of the mammary duct.	They form bundles of varying thickness and length, associated with other macromolecules of the ECM, which determines their architectural structure. This structure allows the mammary epithelium to be supported during pregnancy and lactation, providing the elastic capacity that enables the tissue to return to its original shape after stretching.	[44,46,47]
	Collagen III	This forms structures with characteristics of fibrillar collagen I and is a homotrimer consisting of a single α chain.		
	Collagen V	This forms structures with characteristics of fibrillar collagen I. It is a heterotrimer composed of three different α chains.		
	FN	This is a dimeric glycoprotein (~500 kDa) that mediates cell adhesion, migration, proliferation, and branching morphogenesis.	It is a precursor of the fibril that interacts with other components of the ECM and organizes the interstitial matrix, allowing for the attachment of breast tissue cells.	[44,48,49]
	TN	Glycoproteins have five members: TN-C, TN-R, TN-W, TN-X, and TN-Y. In the mammary gland, TN-C is transiently expressed in the dense stroma surrounding the budding epithelium during embryogenesis.	TN-X maintains tissue elasticity during lactation.	[44,50]
	Sparc	It is a small glycoprotein of 32 kDa.	It is upregulated in mammary gland development during the transition from lactation to postpartum involution, and increased collagen and FN levels correlate with this.	[44,51]
	Laminins (LN, Ln, Lm, or Lam)	The primary protein membrane comprises three polypeptide chains: α, β, and γ. In the mammary gland, LN-111 and LN-332 are abundant at the level of the basement membrane.	They are responsible for acinar formation and induction of contact with epithelial cells. They stimulate milk secretion.	[50]
	Decoration	This is a decorin core protein (~38 kDa) linked to a single chain of CS or DS.	It controls the spatial alignment of collagen fibers in the stroma. It is crucial for the proper organization of fibrillar collagen.	[44,52]
	Biglycan	It comprises a 38 kDa core protein covalently linked to two GAG chains (chondroitin sulfate and/or dermatan sulfate) with an overall MW of 150–240 kDa.	It plays a role in inducing the elastic properties of the gland during periods of expansion.	[44,53]
Fibrous connective tissue	Elastic fibers	They have components such as elastin, fibulins, and proteoglycans associated with microfibrils, forming elastic fibers.	It provides structural support and elasticity to various tissues during lactation.	[44]

**Abbreviations:** CS, Chondroitin sulfate; DS, Dermatan sulfate; ECM, Extracellular matrix; FN, Fibronectin; GAG, Glycosaminoglycan; MW, Molecular weight; TN, Tenascin.

**Table 2 bioengineering-12-00501-t002:** Comparative overview of mammary gland ECM components in bovines and small ruminants.

Characteristic	Bovines	Small Ruminants (Sheep/Goats)	References
Predominant collagen type	Type I collagen is more abundant, contributing to greater tissue rigidity.	Higher proportion of type III collagen, favoring elasticity.	[55,56,57]
FN distribution	More abundant in the parenchyma than in the mammary fat pad.	Similar, but with less quantitative detail in studies.	[56,58,59]
LN distribution	Present in the parenchymal stroma, associated with epithelial organization.	Similar distribution, with analogous structural function.	[56,59]
Elastic fibers	Not specifically emphasized.	Significant presence, associated with tissue elasticity.	[57,59,60]
GAG composition	Lower relative proportion of hydrating GAGs.	A higher presence of chondroitin and heparan sulfate contributes to hydration.	[57,58]
ECM density	High fibrillar density and greater mechanical resistance.	Lower density and a more flexible matrix.	[1,57]
MMP activity	Moderate activity and slower remodeling.	High activity, facilitating tissue regeneration.	[44,60]

**Abbreviations:** ECM, Extracellular matrix; FN, Fibronectin; GAG, Glycosaminoglycan; LN, Laminin; MMP, Metalloproteinase.

**Table 3 bioengineering-12-00501-t003:** Primary decellularization methods, highlighting their advantages and disadvantages.

Methods of Decellularization		Advantages	Disadvantages	Reference
Physical	Flash freezing	This technique is considered safe because it does not produce residual chemicals and has minimal impact on tissue structure and biochemical composition after decellularization.	Damage or rupture of the ECM due to extremely low temperatures.	[73,79,80,81]
	Mechanical force	Cell rupture followed by washing to remove cellular material.	The application of pressure compromises the integrity of the ECM.	[75,80]
	Mechanical agitation	Increases exposure to chemical reagents for cell removal.	Damage to the ECM in cases of agitation or excessive sonication.	[73,75]
	Sonication	Facilitates the penetration of chemical detergents and accelerates the removal of cellular debris.	Potential damage to cell membranes and the ECM due to cavitation.	[82]
Chemical	Ionic detergents (SDS)	SDS solubilizes cytoplasmic and nuclear membranes, inducing cell lysis. In the ECM, SDS removes residual cytoplasmic and cellular proteins.	SDS concentrations exceeding 10 μg/mg dry weight can induce cytotoxicity due to the difficulty of removing SDS from decellularized tissues. As it forms strong hydrophobic bonds with ECM proteins, it may disrupt the native tissue structure, deplete GAGs, and damage collagen and other structural proteins.	[73,83]
	Nonionic detergents	Triton X-100 disrupts lipid–lipid and lipid–protein hydrophobic interactions while preserving protein–protein interactions.	Its efficiency varies by tissue type, yielding mixed results regarding ECM integrity and may deplete GAGs.	[82,83]
	Alkaline and acidic agents	These agents solubilize cytoplasmic components and destroy nucleic acids.	They also result in a significant loss of GAGs from the ECM.	[83,84]
	Zwitterionic detergents	Compounds such as CHAPS exhibit properties of both ionic and nonionic detergents, facilitating cell removal and ECM disruption similar to Triton X-100.	It may damage ECM proteins depending on the tissue and concentration used.	[81,85]
	Hypotonic/hypertonic solutions	Induces cell lysis by osmotic shock.	Ineffective in removing residual cell contents.	[83,86]
	EDTA/EGTA	Disrupts cell–ECM adhesions by chelating divalent metal ions.	It is often used with enzymatic methods, such as trypsin digestion, and has a limited impact when used alone.	[82,87]
Enzymatic	Trypsin	Removes specific cell proteins, facilitating decellularization.	Prolonged exposure can disrupt the structure of the ECM, removing essential components such as LN, FN, elastin, and GAGs.	[81,88]
	Endonucleases	Enzymes catalyze the hydrolysis of internal bonds in ribonucleotide and deoxyribonucleotide chains.	They can complicate the removal of intact cells.	[87,88,89]
	Exonucleases	These enzymes catalyze the hydrolysis of terminal bonds in ribonucleotide and deoxyribonucleotide chains.	Limited impact when used alone.	[81,90]

**Abbreviations:** CHAPS, 3Cholamidopropyl dimethylammonio-1propanesulfonate; ECM, Extracellular matrix; EDTA, Ethylenediaminetetraacetic acid; EGTA, Ethylene glycol-bis(β)-aminoethyl ether; FN, Fibronectin; GAG, Glycosaminoglycan; LN, Laminin; SDS, Sodium dodecyl sulfate.

**Table 4 bioengineering-12-00501-t004:** Comparison of cell types for tissue engineering: definitions, advantages, disadvantages, and applications.

Cellular Type	Definition	Advantages	Disadvantages	Applications	Reference
Fetal and Adult Cells	Fetal cells: maintain phenotypic markers and spatial organization when cultured on biological scaffolds. Adult cells: include renal/alveolar epithelial cells and fibroblasts, often obtained via biopsy.	Fetal cells: show promising functional capabilities in lung, liver, and kidney scaffolds. Adult cells: ease of acquisition via biopsy.	Fetal cells are unsuitable for clinical applications; adult cells have low proliferative capacity and limited scalability for organ repopulation.	Fetal cells recellularize scaffolds, such as rat lung, liver, and kidney; adult cells are used for kidney and lung recellularization, but are limited by low proliferation.	[108,109]
ESCs	Pluripotent stem cells can expand in vitro and differentiate into multiple lineages.	High proliferative capacity; ability to differentiate into multiple lineages; influence cell differentiation in organ matrices.	Ethical concerns regarding the source; potential for teratoma formation (tumorigenicity); risk of uncontrolled differentiation.	Widely used in tissue engineering studies for the recellularization of organ scaffolds.	[75,109,110]
MSCs	Multipotent stem cells isolated from bone marrow or adipose tissue can differentiate into various cell types and support tissue repair.	Robust proliferation in culture; differentiation into multiple lineages; secretion of cytokines and chemokines for tissue repair; provision of stromal support.	Differentiation may be inconsistent across different scaffolds. Require precise culture conditions for lineage-specific differentiation.	Hepatic dECM: accelerates differentiation into hepatocytes; cardiac dECM: differentiates into cardiomyocytes under stimulation; pulmonary dECM: differentiates into epithelial lineages.	[110,111,112,113,114,115]
iPSCs	Stem cells are generated by reprogramming somatic cells to express pluripotency genes, mimicking ESCs.	Adhere to and proliferate on dECM; express alveolar markers in pulmonary scaffolds; and have pluripotent characteristics similar to ESCs.	Lower adhesion in specific scaffolds (e.g., cardiac dECM); risk of genetic instability; labor-intensive reprogramming process.	Pulmonary dECM: promotes adhesion and proliferation. Cardiac dECM: shows lower adhesion than MSCs, requiring further research to optimize use.	[110,115]

**Abbreviations:** dECM, Decellularized extracellular matrix; ESCs, Embryonic stem cells; iPSCs, Induced pluripotent stem cells; MSCs, Mesenchymal stem cells.

**Table 5 bioengineering-12-00501-t005:** Applications of decellularized and recellularized mammary glands.

Species Studied	Cell Type Recellularized	Results	Reference
Bovine	Sheep skin cells	The new material was a biological scaffold for in vitro skin cell culture.	[126]
Rat and Human	Normal mammary and breast cancer cells	The study described a novel mammary-specific culture protocol that combines a self-gelling hydrogel comprised solely of an ECM from decellularized rat or human breast tissue with a 3D bioprinting platform.	[127]
Bovine	No cells were used	A series of in vitro tests demonstrated the consistency and potential of this approach for decellularized xenogenic scaffolds, a concept that had not been explored before.	[128]
Bovine	Endothelial cells	Potential grafts for the treatment of acute ischemia were developed.	[129]
Human	Adipose stem cells	The study illustrated the potential of regenerative medicine in terms of mammary gland reconstruction to restore breast physiology and morphology damaged by mastectomy.	[26]
Minipig	No cells were used	SGBTR regenerates soft tissue by implanting additively manufactured bioresorbable scaffolds filled with autologous fat grafts.	[130]

**Abbreviations:** 3D, Three-dimensional; ECM, Extracellular matrix; SGBTR, Scaffold-guided breast tissue regeneration.

**Table 6 bioengineering-12-00501-t006:** Studies utilizing 3D printing for recellularizing mammary gland tissues in different species.

Species Studied	Main Results	Reference
Multiple eutherian mammals and a marsupial (gray short-tailed opossum)	Successfully created next-generation 3D mammary gland organoids from eight eutherian mammals and the first branched organoid of a marsupial mammary gland, providing a model for studying mammary gland evolution and development.	[141]
Human mammary epithelial cells	Developed a 3D bioprinting protocol for normal and cancerous mammary epithelial cells into a branched Y shape, facilitating the study of cell positioning in regulating proliferation and invasion.	[94,142]
Mouse mammary epithelial cells	Detailed the use of a 3D bioprinting platform to control the formation of organoids through the “self-assembly” of mammary epithelial cells, enabling consistent and reproducible cultures of large-scale 3D mammary epithelial tissues.	[143]
Porcine breast tissue	Developed a method for decellularizing and delipidating porcine breast tissue compatible with hydrogel formation, advancing the development of tissue-engineered breast models.	[94,144]
Canine mammary gland tumors	Utilized 3D culture methods to model canine mammary gland tumors, providing insights into tumor biology and potential therapeutic approaches.	[145]
Various organoids	Discussed organoid bioprinting approaches that control the 3D arrangement of organoids, contributing to the development of functional tissue for regenerative medicine.	[138,146]

**Abbreviations:** 3D, Three-dimensional.

## Data Availability

No new data were created or analyzed in this study. Data sharing is not applicable to this article.

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
