# Peer review of "A Review on Bioengineering the Bovine Mammary Gland: The Role of the Extracellular Matrix and Reconstruction Prospects"

_bioengineering, 2025, doi:10.3390/bioengineering12050501_

Round 1
Reviewer 1 Report
Comments and Suggestions for Authors
Júnior et al., reviewed the study entitled as “Bioengineering the Bovine Mammary Gland: The Role of the Extracellular Matrix and Reconstruction Prospects” is a good, organized review. They provided literature on biomaterials and tissue engineering, and giving overview of the ECM of the bovine mammary gland and advances in its bioengineering, with a focus on regenerative medicine for bovines. However, I have some suggestive comments for the improvement of this manuscript.
Some minor comments:
-The abbreviations appearing for the first time need to be given full names.
-please provide a table containing all the abbreviations used in the manuscript.
-please add review in your title.
-the methodology of the review has not been described in abstract.
-please mention the from which search databases (e.g., PubMed, Scopus, Cochrane library and Google Scholar).
-please also mention the duration of data collection (like, 1980 to 2018 etc.)
Fig.1A, why both mammary glands are looking different.
- Fig.1E, in figure please write full “Recellularization”.
- Fig.1H, PLEASE correct the spelling of “Pathology”.
- Fig.2B, as Fig.2c, 2b has not shown which area of bovine udder had been enlarged.
- Fig.2, caption 2ABC please capitalized the sentence starting word.
-Please also link your manuscript with treatment of mastitis.
- Introduction section needs improvements in terms of basic story and recent literature, so many old refences are there. Some recent and suitable references can be added to support this are suggested below.
Meçaj R, Muça G, Koleci X, Sulçe M, Turmalaj L, Zalla P, Koni A and Tafaj M, 2023. Bovine environmental mastitis and their control: an overview. International Journal of Agriculture and Biosciences 12(4) 216-221. https://doi.org/10.47248/journal.ijab/2023.067
Yang B: Lipoteichoic acid disrupts mammary epithelial barrier integrity by altering expression of occludin and zonula occluden (ZO)-1. Kafkas Univ Vet Fak Derg, 29 (4): 423-428, 2023. DOI: 10.9775/kvfd.2023.29103
Mehmood S and M Ashraf, 2023. Antimicrobial resistance and virulence determinants of E. coli in bovine clinical mastitis in dairy farms. Continental Vet J, 3(1):54-59. http://dx.doi.org/10.71081/cvj/2023.008
Yue S, Qian J, Du J, Liu X, Xu H, Liu H, Zhang J, Chen X. 2023. Heat stress negatively influence mammary blood flow, mammary uptake of amino acids and milk amino acids profile of lactating holstein dairy cows. Pak Vet J, 43(1): 73-78. http://dx.doi.org/10.29261/pakvetj/2023.002
Muhammad G, S Parveen, I Rashid and S Rashid, 2023. SWOT Analysis of a rural and peri-urban outreach mastitis control program. Continental Vet J, 3(2):103-109. http://dx.doi.org/10.71081/cvj/2023.025
Ghazvineh N, Mokhtari A, Ghorbanpoor Najaf Abadi M, Kadivar A, Shahrokh Shahraki S: Molecular detection of selective virulence factors of Mycoplasma bovis local isolates involved in bovine mastitis. Kafkas Univ Vet Fak Derg, 30 (5): 631-639, 2024. DOI: 10.9775/kvfd.2024.32118
Hayajneh FMF, Ahmed Z, Khatoon A, Saleemi MK Arshad MI and Gul ST, 2024. Epidemiological investigations of Mycoplasma bovis-associated mastitis in dairy animals along with analysis of interleukin-6 (IL-6) as a potential diagnostic marker. International Journal of Veterinary Science 13(1): 120-126. https://doi.org/10.47278/journal.ijvs/2023.072
-Finally, I advise passing the whole manuscript for a rigorous English check.
Comments on the Quality of English Languageminor revision.
Author Response
RESPONSE TO REVIEWERS' COMMENTS
Manuscript number: bioengineering-3582590 ― Bioengineering (MDPI)
"A review on Bioengineering the Bovine Mammary Gland: The Role of the Extracellular Matrix and Reconstruction Prospects"
The authors of this document wish to express their deepest gratitude to the Editor-in-Chief and the Reviewer for their thorough and insightful evaluation of our manuscript. Their expert feedback has been invaluable in enhancing the quality of our work. We have carefully considered and diligently implemented each suggestion, significantly improving the manuscript. We have made substantial revisions to address the points raised. These noteworthy changes are marked mainly with YELLOW-highlighted text throughout the document for ease of reference. A note will be provided for the referee's attention for corrections highlighted in a different color. Additionally, we have prepared a detailed and comprehensive response to each comment and suggestion. This response is organized in a "point-by-point" format below, ensuring that every concern has been thoroughly addressed and explained. We sincerely appreciate the time and effort invested by the Editor-in-Chief and the Reviewer, and we believe their contributions have significantly strengthened the final version of our manuscript.
REVIEWER #1
General comment
Júnior et al., reviewed the study entitled as “Bioengineering the Bovine Mammary Gland: The Role of the Extracellular Matrix and Reconstruction Prospects” is a good, organized review. They provided literature on biomaterials and tissue engineering, and giving overview of the ECM of the bovine mammary gland and advances in its bioengineering, with a focus on regenerative medicine for bovines. However, I have some suggestive comments for the improvement of this manuscript.
General response
Dear Erudite Reviewer, thank you for taking the time to revise our manuscript and for providing us with the opportunity to improve based on your valuable comments and suggestions. After addressing all your comments and suggestions regarding our manuscript text, we are confident that a significantly improved manuscript version has emerged. We are excited to resubmit the modified version for your kind perusal and reevaluation. Thank you for your brilliant insights, essential contributions, and feedback. You do have an eye for improvement. As a gesture of our utmost respect for you, we would like to provide you with a detailed and comprehensive point-by-point response to your comments below. Thank you once again for your time and patience in revising our article.
Comment 1
The abbreviations appearing for the first time need to be given full names.
Response
Thank you for your comment. To address this point, we have carefully reviewed the entire manuscript and ensured that all abbreviations are spelled out the first time they appear in the text. Additionally, we have included a table listing all abbreviations and their corresponding full terms to enhance clarity for the readers.
Comment 2
Please provide a table containing all the abbreviations used in the manuscript.
Response
After the first observation, we included a table. This adjustment may improve the quality of the manuscript.
Comment 3
Please add review in your title.
Response
Thank you for your suggestion. We revised the manuscript title to include the word "review", making the nature of the article more transparent to readers.
Comment 4
The methodology of the review has not been described in abstract.
Response
Thank you for your observation. We have now included a brief description of the methodology in the abstract, specifically summarizing the databases used, the time frame considered, and the criteria for reference selection. This addition appears between lines 39 and 43. This improves the transparency and completeness of the abstract.
Comment 5
Please mention the from which search databases (e.g., PubMed, Scopus, Cochrane library and Google Scholar).
Response
Thank you for your comment. The search databases used for reference collection, PubMed, Scielo, and Google Scholar, are now clearly mentioned in the manuscript between lines 39 and 40.
Comment 6
Please also mention the duration of data collection (like, 1980 to 2018 etc.)
Response
Thank you for your valuable suggestion. We have now included the duration of the data collection in the manuscript, which spans from 2002 to 2025. This information is provided in line 42 to clarify the time frame for the references included in the review. with particular attention to literature published in the last 10 years, to provide a contemporary perspective.
Comment 7
Fig.1A, why both mammary glands are looking different.
- Fig.1E, in figure please write full “Recellularization”.
- Fig.1H, PLEASE correct the spelling of “Pathology”.
Response
Thank you for your detailed observations.
- In Fig.1A, the left side shows cut alveoli, while the right side shows intact, whole alveoli, which accounts for the difference in appearance.
- The full term, Recellularization, we did.
- The spelling of Pathology has been corrected
Comment 8
Fig.2B, as Fig.2c, 2b has not shown which area of bovine udder had been enlarged.
- Fig.2, caption 2ABC please capitalized the sentence starting word.
Response
Thank you for your observations.
- In Fig. 2 B, we have clarified in the caption which area of the bovine udder has been enlarged, as suggested.
- The starting words of the sentences in the captions for Fig.2A, Fig.2B, and Fig.2C have been capitalized for consistency with standard formatting.
Comment 9
Please also link your manuscript with treatment of mastitis.
Response
Thank you for your suggestion. We have now included a section reviewing the treatment of mastitis in the manuscript, which spans from line 308 to line 325. This addition links the manuscript to the treatment approaches for mastitis, as requested.
Comment 10
Introduction section needs improvements in terms of basic story and recent literature, so many old refences are there. Some recent and suitable references can be added to support this are suggested below.
Meçaj R, Muça G, Koleci X, Sulçe M, Turmalaj L, Zalla P, Koni A and Tafaj M, 2023. Bovine environmental mastitis and their control: an overview. International Journal of Agriculture and Biosciences 12(4) 216-221. https://doi.org/10.47248/journal.ijab/2023.067
Yang B: Lipoteichoic acid disrupts mammary epithelial barrier integrity by altering expression of occludin and zonula occluden (ZO)-1. Kafkas Univ Vet Fak Derg, 29 (4): 423-428, 2023. DOI: 10.9775/kvfd.2023.29103
Mehmood S and M Ashraf, 2023. Antimicrobial resistance and virulence determinants of E. coli in bovine clinical mastitis in dairy farms. Continental Vet J, 3(1):54-59. http://dx.doi.org/10.71081/cvj/2023.008
Yue S, Qian J, Du J, Liu X, Xu H, Liu H, Zhang J, Chen X. 2023. Heat stress negatively influence mammary blood flow, mammary uptake of amino acids and milk amino acids profile of lactating holstein dairy cows. Pak Vet J, 43(1): 73-78. http://dx.doi.org/10.29261/pakvetj/2023.002
Muhammad G, S Parveen, I Rashid and S Rashid, 2023. SWOT Analysis of a rural and peri-urban outreach mastitis control program. Continental Vet J, 3(2):103-109. http://dx.doi.org/10.71081/cvj/2023.025
Ghazvineh N, Mokhtari A, Ghorbanpoor Najaf Abadi M, Kadivar A, Shahrokh Shahraki S: Molecular detection of selective virulence factors of Mycoplasma bovis local isolates involved in bovine mastitis. Kafkas Univ Vet Fak Derg, 30 (5): 631-639, 2024. DOI: 10.9775/kvfd.2024.32118
Hayajneh FMF, Ahmed Z, Khatoon A, Saleemi MK Arshad MI and Gul ST, 2024. Epidemiological investigations of Mycoplasma bovis-associated mastitis in dairy animals along with analysis of interleukin-6 (IL-6) as a potential diagnostic marker. International Journal of Veterinary Science 13(1): 120-126. https://doi.org/10.47278/journal.ijvs/2023.072
Response
Thank you for your comment. We have revised the Introduction by removing outdated references and incorporating more recent and relevant literature. The section has been improved, adding additional information between lines 55 and 75.
Comment 11
Finally, I advise passing the whole manuscript for a rigorous English check.
Response
Dear Erudite Reviewer, thank you for this important suggestion. While addressing your precious comments and recommendations, we asked for a native English speaker to read and polish our manuscript accordingly. We believe our manuscript has been significantly improved regarding the English language. Thank you for everything!
I, the corresponding author of the manuscript "A review on Bioengineering the Bovine Mammary Gland: The Role of the Extracellular Matrix and Reconstruction Prospects" under the assigned ID bioengineering-3582590, on behalf of my coauthors, once again extend my heartfelt gratitude to the knowledgeable Editor-in-Chief and reviewers for their time and expertise in revising our manuscript. After we addressed their constructive and refined feedback and suggestions, a significantly improved manuscript version emerged. Undoubtedly, their insightful suggestions and feedback have significantly enhanced the quality of our manuscript. We respectfully are at the disposal of the Editor-in-Chief and the Reviewer to address any additional suggestions regarding our publication. If you are satisfied with our newly refined and significantly improved version, we look forward to the acceptance of our article for publication in this prestigious journal, Bioengineering. Thank you once again for your time and expertise.
Reviewer 2 Report
Comments and Suggestions for Authors
The authors have presented a review manuscript regarding the significance of the extracellular matrix in the bovine mammary gland.
Points to consider for correction.
- Although this is not a systematic review, the authors must include a section regarding the collection of references. How did they choose references for inclusion in the review? I noticed that there are many references with similar authorship, whilst there are other important references missing. Which databases were employed? Which is the timespan, during which references were published? These points should be addressed, as at the moment the manuscript is a bit biased and also that way future readers will lack information.
- The first two paragraphs of the Introduction are totally irrelevant and must be deleted.
- The manuscript does not offer any information regarding the role of ECM in the cellular defence mechanisms of the mammary gland. This is a serious omission and must be added.
- A new section must be included to compare ECM in cattle with similar findings in other domestic ruminant species.
- The number of references is small and really does not cover fully the topic. For such a complex and interesting topic, I expect at least 200 references to be included in the manuscript.
- Please totally correct this section to include a strong takeaway message.
Overall. Extensive corrections by taking into account the above and re-evaluation.
Author Response
RESPONSE TO REVIEWERS' COMMENTS
Manuscript number: bioengineering-3582590 ― Bioengineering (MDPI)
"A review on Bioengineering the Bovine Mammary Gland: The Role of the Extracellular Matrix and Reconstruction Prospects"
The authors of this document wish to express their deepest gratitude to the Editor-in-Chief and the Reviewer for their thorough and insightful evaluation of our manuscript. Their expert feedback has been invaluable in enhancing the quality of our work. We have carefully considered and diligently implemented each suggestion, significantly improving the manuscript. We have made substantial revisions to address the points raised. These noteworthy changes are marked mainly with YELLOW-highlighted text throughout the document for ease of reference. A note will be provided for the referee's attention for corrections highlighted in a different color. Additionally, we have prepared a detailed and comprehensive response to each comment and suggestion. This response is organized in a "point-by-point" format below, ensuring that every concern has been thoroughly addressed and explained. We sincerely appreciate the time and effort invested by the Editor-in-Chief and the Reviewer, and we believe their contributions have significantly strengthened the final version of our manuscript.
REVIEWER #1
General comment
The authors have presented a review manuscript regarding the significance of the extracellular matrix in the bovine mammary gland.
General response
Dear Erudite Reviewer, thank you for taking the time to revise our manuscript and for providing us with the opportunity to improve based on your valuable comments and suggestions. After addressing all your comments and suggestions regarding our manuscript text, we are confident that a significantly improved manuscript version has emerged. We are excited to resubmit the modified version for your kind perusal and reevaluation. Thank you for your brilliant insights, essential contributions, and feedback. You do have an eye for improvement. As a gesture of our utmost respect for you, we would like to provide you with a detailed and comprehensive point-by-point response to your comments below. Thank you once again for your time and patience in revising our article.
Comment 1
Although this is not a systematic review, the authors must include a section regarding the collection of references. How did they choose references for inclusion in the review? I noticed that there are many references with similar authorship, whilst there are other important references missing. Which databases were employed? Which is the timespan, during which references were published? These points should be addressed, as at the moment the manuscript is a bit biased and also that way future readers will lack information.
Response
We appreciate the reviewer’s valuable observations. In response to similar comments from the first reviewer, we have clarified our methodology and reference selection process in the Introduction and abstract lines 39–43.
Comment 2
The first two paragraphs of the Introduction are totally irrelevant and must be deleted.
Response
Thank you for your constructive feedback. As suggested, the first two paragraphs of the Introduction have been removed. Additionally, the introduction has been revised and improved between lines 55 and 75 to ensure better coherence and relevance to the topic.
Comment 3
The manuscript does not offer any information regarding the role of ECM in the cellular defence mechanisms of the mammary gland. This is a serious omission and must be added.
Response
Thank you for your valuable suggestion. In response, we have now included a review of The Role of the Extracellular Matrix (ECM) in the Immune Response During Mastitis, which can be found between lines 264 and 279. This addition addresses the role of ECM in cellular defense mechanisms of the mammary gland, as requested.
Comment 4
A new section must be included to compare ECM in cattle with similar findings in other domestic ruminant species.
Response
Thank you for your suggestion. As requested, we have included a new section comparing the ECM in cattle with similar findings in other domestic ruminant species. This section has been added between lines 268 and 279 to address the comparison and provide a broader perspective on the topic.
Comment 5
The number of references is small and really does not cover fully the topic. For such a complex and interesting topic, I expect at least 200 references to be included in the manuscript.
Response
Thank you for your valuable feedback. While we could not reach the minimum number of references requested, we have increased the number of references significantly. We focused on including recent and highly relevant articles that provide substantial and valuable information on the topic. These updates enhance the manuscript's depth and comprehensiveness.
Comment 6
Please totally correct this section to include a strong takeaway message.
Response
Thank you for your suggestion. As requested, the new challenges and conclusions section has been completely rewritten to include a stronger and clearer takeaway message.
Comment 7
Overall. Extensive corrections by taking into account the above and re-evaluation.
Response
Dear Erudite Reviewer, thank you for this important suggestion. While addressing your precious comments and recommendations, we asked for a native English speaker to read and polish our manuscript accordingly. We believe our manuscript has been significantly improved regarding the English language. Thank you for everything!
I, the corresponding author of the manuscript "A review on Bioengineering the Bovine Mammary Gland: The Role of the Extracellular Matrix and Reconstruction Prospects" under the assigned ID bioengineering-3582590, on behalf of my coauthors, once again extend my heartfelt gratitude to the knowledgeable Editor-in-Chief and reviewers for their time and expertise in revising our manuscript. After we addressed their constructive and refined feedback and suggestions, a significantly improved manuscript version emerged. Undoubtedly, their insightful suggestions and feedback have significantly enhanced the quality of our manuscript. We respectfully are at the disposal of the Editor-in-Chief and the Reviewer to address any additional suggestions regarding our publication. If you are satisfied with our newly refined and significantly improved version, we look forward to the acceptance of our article for publication in this prestigious journal, Bioengineering. Thank you once again for your time and expertise.
Round 2
Reviewer 2 Report
Comments and Suggestions for Authors
All the issues were addressed. No further comments.